# Effects of Ecological Restoration and Climate Change on Herbaceous and Arboreal Phenology

**DOI:** 10.3390/plants12223913

**Published:** 2023-11-20

**Authors:** Zixuan Yuan, Yiben Cheng, Lina Mi, Jin Xie, Jiaju Xi, Yiru Mao, Siqi Xu, Zhengze Wang, Saiqi Wang

**Affiliations:** 1Breeding Base for State Key Lab. of Land Degradation and Ecological Restoration in Northwestern China, Yinchuan 750021, China; yzx0124@bjfu.edu.cn; 2Key Lab. of Restoration and Reconstruction of Degraded Ecosystems in Northwestern China of Ministry of Education, Ningxia University, Yinchuan 750021, China; 3School of Soil and Water Conservation, Beijing Forestry University, Beijing 100083, China; 18253195806@163.com (Y.M.); x725866@163.com (S.X.); 13995340138@163.com (Z.W.); 15615451348@163.com (S.W.); 4National Meteorological Centre, China Meteorological Administration, Beijing 100081, China; xiej@cma.gov.cn; 5Department of Remote Sensing and Mapping, Space Star Technology Co., Ltd., Beijing 100086, China; xijj_cast503@sina.com

**Keywords:** climate change, NDVI, vegetation phenology, Horqin Sandy Land, meteorological factors

## Abstract

With global climate change, changes in vegetation phenology have become increasingly evident. Horqin Sandy Land is located near the eastern part of the West Liaohe River. It is the largest sandy land in China and its ecological environment is fragile. Investigating the changes in vegetation phenology in these sandy areas and determining the relationship between vegetation phenology and meteorological factors are of great importance for predicting the impacts of future climate change and understanding the response mechanisms of ecosystems. In this study, we used the time series of the Normalized Difference Vegetation Index (NDVI) from 2000 to 2021 and extracted the vegetation phenology in the Horqin Sandy Land using high-order curve fitting methods, including the start date of the growing season (SOS), the end date of the growing season (EOS), and the length of the growing season (LOS). We analyzed their temporal variation and used partial correlation analysis to determine their relationship with meteorological factors (temperature and precipitation). In addition, we compared the phenology and microclimate of forest and grassland within the study area. In the Horqin Sandy Land, the vegetation SOS was concentrated between the 115th and 150th day, the EOS was concentrated between the 260th and 305th day, and the LOS ranged from 125 to 190 days. Over the past 22 years, the SOS, EOS, and LOS of vegetation in the Horqin Sandy Land showed trends of delay, shift, and extension, with rates of change of 0.82 d/10a, 5.82 d/10a, and 5.00 d/10a, respectively. The start date of the growing season in the Horqin Sandy Land was mainly influenced by precipitation in April of the current year, while the end date was mainly influenced by precipitation in August of the current year. Overall, the SOS in the forested areas of the Horqin Sandy Land was slightly later than in the grasslands, but the EOS in the forested areas was significantly later than in the grasslands, resulting in a longer LOS in the forests. In addition, annual precipitation and the rate of precipitation increase were higher in the forested areas than in the grasslands, but soil temperature was higher in the grasslands than in the forests. Vegetation phenology in the Horqin Sandy Land has undergone significant changes, mainly manifested in the delayed end date of the growing season, the extended length of the growing season, and the differences between forest and grassland. This indicates that climate change has indeed affected phenological changes and provides a theoretical basis for subsequent ecological restoration and desertification prevention efforts in the region.

## 1. Introduction

Over the past century, the global warming trend has become increasingly evident, with the average surface temperature increasing by 0.74 °C, and the rate of temperature increase reaching 0.13 °C/10a over the past 50 years [1,2]. Climate change has had significant impacts on the natural environment and human production and activities [3], which are particularly evident in the changes in vegetation phenology [4,5]. Vegetation phenology refers to the natural phenomena of long-term variations in vegetation due to the climate and its natural environment [6], occurring on an annual cycle [7], such as budding, leaf unfolding, flowering, fruiting, leaf shedding, and dormancy [8]. Vegetation phenology is a highly sensitive indicator of global climate change, objectively reflecting the response and adaptability of plant growth to external climatic and ecological conditions [9,10,11]. Therefore, phenological research has become an important part of climate change studies and is crucial for predicting the impacts of future climate change and understanding the response mechanisms of ecosystems.

The effect of ecological restoration projects on vegetation phenology in desert areas is an important research topic in the field of ecology [12,13,14]. The ecological environment in desert regions is fragile [15], with low vegetation cover, and phenological events are significantly influenced by climatic factors and human activities [16,17]. To improve the ecological environment of desert regions, various ecological restoration projects have been widely implemented, including vegetation restoration, soil and water conservation, afforestation, and grassland rehabilitation [18]. Ecological restoration projects can alter the water conditions and soil quality in desert regions [19,20,21], thereby affecting plant growth and development [22]. Studies have shown that these restoration activities can advance or delay the phenological phases of vegetation [23,24,25], making it more adaptable to local climatic conditions. Vegetation in desert areas often has a short phenological period [26], but ecological restoration projects can extend the growing season of plants by increasing land temperature and local annual precipitation [14,27,28], resulting in a longer phenological phase length [29]. The harsh climatic conditions in desert regions make vegetation phenology vulnerable to climate variations and human disturbance. Ecological restoration projects can improve the stability of vegetation phenology, making it more resilient to adverse climatic conditions [30].

The identification of vegetation phenology can be achieved by controlled experiments, field observations, remote sensing monitoring, and model simulations [14]. Among these methods, remote sensing monitoring of vegetation phenology, which is based on the curve changes of satellite remote sensing of vegetation index time series [31], has the advantages of longer time series and broader coverage compared with other methods, making it the main means of phenological observation. In recent years, many scholars have conducted research on vegetation phenological changes using remote sensing monitoring methods [32]. Research shows that from 1982 to 2014, 13.5% of the study areas in northern China showed a significant trend of earlier start dates of the growing season (SOS), and 23.1% of the study areas showed a significant trend of delayed end dates of the growing season (EOS) [33]. The research by Zheng Guanheng et al. shows that from 2001 to 2020, significant changes occurred in vegetation phenology in the permafrost region of the Three-River Source area, with a 6.3 d/10a lengthening of the growing season mainly attributed to a 4.9 d/10a advancing of the start date of the growing season (SOS) [34].

The Horqin Sandy Land is located near the eastern part of the West Liao River and is the largest sandy land in China [35]. It is located in the transition zone between northern agriculture and animal husbandry, characterized by semi-arid and semi-humid sandy-alkali soil, with a fragile ecological environment. Therefore, studying the changes in vegetation phenology and its response to climatic factors in this region is important for relevant research on typical vegetation in sandy areas in China [36]. Based on this, this study focuses on the Horqin Sandy Land as the study area and uses remote sensing monitoring methods to explore the dynamic changes in phenology and climate from 2000 to 2021, their relationship, the differences in phenology between forest and grassland areas within the region, and the analysis of the reasons and advantages and disadvantages of the changes of some climate factors. The aim is to provide a basis for the study of vegetation in sandy areas and the research of remote sensing technology on phenology.

## 2. Materials and Methods

### 2.1. Study Area Overview

The Horqin Sandy Land is located in the northeastern part of Chifeng City, Inner Mongolia Autonomous Region [37], China (Figure 1), with a central latitude of approximately 43°08′ N and a longitude of 119°38′ E. The region falls within the temperate semi-arid monsoon climate zone, characterized by continental climate conditions. It experiences dry and windy springs, hot and rainy summers, cool autumns, and cold winters with little snow. The average annual precipitation is 368 mm, and the average annual evaporation is 1690 mm. The average annual temperature ranges from 5 °C to 7.5 °C. The frost period lasts from November to April of the following year, and the frost-free period ranges from 140 to 280 days. The prevailing winds are the north wind and northwest wind, with an average annual wind speed of 3.6 m/s. The frequency of sand-lifting winds with speeds ≥ 5 m/s is 210 to 310 days per year, even reaching up to 330 days. The frequency of strong winds with speeds ≥ 17.2 m/s is 25 to 40 days per year, with sandstorm weather occurring for 10 to 15 days, mainly in the spring [38].

The Horqin Sandy Land has undulating and extensive terrain, with higher elevations in the west and lower elevations in the east, while the central part is relatively lower. The elevation ranges from 120 to 800 m. The dominant soil type is sandy chestnut soil, but due to intense desertification, much of the soil has degraded into wind-blown sand. Dunes cover 53% of the total land area, with mobile dunes accounting for 5.5%, semi-fixed dunes 20%, and fixed dunes 27.5%. The ratio of sand dunes to grassland is approximately 7:3. Plant species representative of mobile dunes include *Caragana microphylla*, *Artemisia ordosica*, *Setaria viridis*, and *Pennisetum centrasiaticum*. In contrast, fixed dunes are dominated by shrubs such as *Calligonum mongolicum*, *Tamarix chinensis*, *Salix cheilophila*, *Artemisia sacrorum*, and *Berchemia polyphylla*.

### 2.2. Data Source

The remote sensing data used in this study were obtained from the Google Earth Engine (GEE) cloud platform, which provides both data collection and powerful spatial analysis capabilities [39]. It supports the acquisition and analysis of long-time series of remote sensing imagery, enabling integrated data retrieval, processing, analysis, and application.

#### 2.2.1. NDVI Data Source

The Normalized Difference Vegetation Index (NDVI) is an essential parameter for reflecting vegetation vigor and nutrient information on the Earth’s surface, allowing the detection, analysis, and assessment of vegetation growth status and cover [40].

The NDVI data used in this study were acquired from the Moderate-resolution Imaging Spectroradiometer (MODIS), a large space-based remote sensing instrument developed by NASA. (Table 1) MODIS is carried on both the Terra and Aqua satellites and is an important instrument in the NASA Earth Observing System program for observing global biological and physical processes. Specifically, the MOD09GA_006_NDVI data have a resolution of 500 m, providing NDVI values for every day of the year. For this research, we selected MODIS imagery for the study area from 2000 to 2021. Using the GEE platform, the data were formatted, mosaicked, and clipped. Based on the MODIS images, the NDVI values of each day are calculated on the platform, and the data is denoised by S-G filtering. Additionally, we divided the study area into 13 regions (Figure 2) and randomly selected three plots each of forest and grassland within each region, and repeated the above operations to obtain relevant data for forest and grassland areas within the study area.

#### 2.2.2. Temperature Data Source

The temperature data used in this study, including air temperature and ground temperature, were obtained from ERA5, the fifth-generation atmospheric reanalysis by the European Centre for Medium-Range Weather Forecasts (ECMWF) [41]. Reanalysis combines model data with observations from around the world, resulting in a comprehensive and consistent global dataset.

#### 2.2.3. Precipitation Data Source

The precipitation data used in this study were obtained from the CHIRPS Daily dataset (Climate Hazards Group InfraRed Precipitation with Station Data). CHIRPS is a dataset that has recorded global precipitation from 1981 to the present. It combines 0.05° resolution satellite imagery with in situ station data to produce gridded precipitation time series for trend analysis and seasonal drought monitoring.

### 2.3. Data Processing

#### 2.3.1. Vegetation Phenology Extraction Method

Normalized Difference Vegetation Index (NDVI) is commonly used for regional-scale vegetation classification and vegetation cover studies. In this study, a high-order curve-fitting method was employed to fit the time series curve of NDVI:(1)$${NDVI}_{t}=a_{0}+a_{1}x_{1}+a_{2}x_{2}+\ldots +a_{n}x_{n}$$
where *x* represents each day of the year; $a_{0}$, $a_{1}$, $a_{2}$, …, $a_{n}$ are the parameters of the high-order curve fitting.

The phenological period was determined using the maximum slope method:(2)$${NDVI}_{ratio(t)}=\frac{{NDVI}_{t+1}{-NDVI}_{t}}{{NDVI}_{t}}$$
where ${NDVI}_{ratio(t)}$ represents the slope of NDVI at time *t*. Considering that most vegetation in the study area stops photosynthesis in November and is influenced by snow cover, making it difficult to observe vegetation growth from remote sensing images, we selected NDVI time series data from April to October for the years 2000 to 2021 for analysis. The date corresponding to the maximum value represents the start of the growing season, and the date corresponding to the minimum value represents the end of the growing season. The difference between the end date and the start date of the growing season represents the length of the growing season.

#### 2.3.2. Trend Analysis Method

A one-dimensional linear regression method was used to analyze the trend of phenological changes from 2000 to 2021.
(3)$$SLOPE=\frac{n\times \sum_{i=1}^{n}i\times T_{i}-\sum_{i=1}^{n}i\times \sum_{i=1}^{n}T_{i}}{n\times \sum_{i=1}^{n}i^{2}-{\left(\sum_{i=1}^{n}i\right)}^{2}}$$
where *n* is the cumulative number of years—in this study, it is 22; the variable *T* represents the phenological parameter data value for the *i*-th year; *SLOPE* represents the slope of the one-dimensional linear regression equation, and a positive or negative slope value indicates a trend of delay or advancement in the phenological data during the study period. A slope value of 0 indicates no significant change in phenological data during the study period.

#### 2.3.3. Partial Correlation Analysis

Vegetation responds to climate change with a lag, which means that vegetation phenological dates can be influenced by meteorological factors from the previous season. Additionally, meteorological factors can have overlapping effects on vegetation. Therefore, analyzing the relationship between monthly temperature precipitation and phenology separately may not be very accurate. In this study, we conducted partial correlation analysis to explore the relationship between vegetation phenology and meteorological factors. For the start date of the growing season, we selected the monthly temperature and precipitation from November of the previous year to April of the current year, as well as the cumulative average temperature and cumulative precipitation from April of the current year backward by 1–5 months, resulting in a total of 22 meteorological factors. Similarly, for the end date of the growing season, we selected monthly temperature and precipitation from April to September of the current year, along with the cumulative average temperature and cumulative precipitation from September of the current year backward by 1–5 months, resulting in a total of 22 meteorological factors. This approach was used to investigate the relationship between vegetation phenology and meteorological factors. Since precipitation and temperature can influence each other, partial correlation analysis was used to determine the relationship between each meteorological element and the start and end dates of the vegetation growing season.

## 3. Results

### 3.1. Vegetation Phenological Changes in the Horqin Sandy Land

Climate change is an important influencing factor on vegetation phenology, and the combined effects of temperature and precipitation are conducive to the growth and development of vegetation. In the Horqin Sandy Land from 2000 to 2021, the start date and end date of the phenological period were mainly concentrated between the 115th and 150th day and the 260th and 305th day, respectively. The start date of the phenological period was mainly in May, with exceptions in April for the years 2009, 2012, 2013, and 2018. The start date of the vegetation phenological period was delayed from the 126th day in 2000 to the 132nd day in 2021, with a change rate of 0.82 days per decade (Figure 3). The end date of the phenological period was mainly in October, except for September 2001. The end date of the vegetation phenological period has been delayed from the 280th day in 2000 to the 301st day in 2021, with a change rate of 5.82 days per decade (Figure 3). The length of the phenological period ranged from 125 to 190 days, changing from 154 days in 2000 to 169 days in 2021, with a change rate of 5.00 days per decade (Figure 3).

### 3.2. Relationship between Vegetation Phenology and Climate Factors in the Horqin Sandy Land

Vegetation phenological changes are correlated with pre-season temperatures. In the Horqin Sandy Land, the start date of the vegetation phenological period is positively correlated with the monthly temperatures from the previous November to the current April and the cumulative average temperatures from April to the preceding 1–5 months, except for a negative correlation with the temperatures in April and the cumulative average temperature for the preceding month (Figure 4). It is also negatively correlated with the monthly precipitation from the previous November to the current April and the cumulative precipitation from April to the previous 1–5 months, except for a positive correlation with the precipitation in the previous December (Figure 4).

The end date of the phenological period is negatively correlated with the monthly temperatures from April to September and the cumulative average temperatures from September to the preceding 1–5 months, except for negative correlations with the temperatures in May, August, and September (Figure 4). It is positively correlated with the monthly precipitation from April to September and the cumulative precipitation from September to the preceding 1–5 months, except for negative correlations with the precipitation in May and June (Figure 4).

Overall, the start date of the phenological period for both forested and grassland areas in the Horqin Sandy Land is positively correlated with temperature factors, with a stronger correlation in grasslands. It is negatively correlated with precipitation factors, and the difference between forests and grasslands is insignificant (Figure 5). The end date of the phenological period for forests is negatively correlated with temperature factors and positively correlated with precipitation factors, while for grasslands, it is positively correlated with both temperature and precipitation factors. The correlation between temperature factors and the end date of the phenological period is stronger in forests than in grasslands, while the correlation with precipitation factors is similar between forests and grasslands (Figure 5). Temperature is the main factor causing the differences in phenology between forests and grasslands.

### 3.3. Changes in Meteorological Factors in the Horqin Sandy Land

In the context of global climate warming, the impact of human activities and climate change on ecosystems is increasingly intensifying. In the Horqin Sandy Land from 2000 to 2021, the annual average NDVI value showed slight fluctuations but exhibited an overall increasing trend. In 2017, 2018, and 2019, the NDVI values were greater than 0.22, with a growth rate of 0.01/10a (Figure 6). The annual average temperature showed a significant upward trend, with a growth rate of 0.29 °C/10a, and it exceeded 8 °C in 2007 and 2019, showing two peaks (Figure 6). Gradual warming has become a prominent feature of climate change in this region. The annual precipitation showed a significant upward trend, with a growth rate of 93.16 mm/10a (Figure 6). An increase in precipitation is another prominent feature of climate change in this area.

From 2000 to 2021, both the ground temperature and precipitation in the forest and grassland areas of the Horqin Sandy Land have shown changes with significant fluctuations, overall displaying an increasing trend. Over the past 22 years, the ground temperature in the forest area has increased from 5.22 °C in 2000 to 6.10 °C in 2021, while in the grassland area, it has increased from 6.84 °C in 2000 to 7.44 °C in 2021 (Table 2). The precipitation in the forest area has increased from 337.25 mm in 2000 to 647.94 mm in 2021, and in the grassland area, it has increased from 343.83 mm in 2000 to 563.70 mm in 2021 (Table 3).

The average ground temperature in the grassland of the Horqin Sandy Land is higher than that in the forest, with a change rate of 0.21 °C/10a for the grassland and 0.31 °C/10a for the forest (Figure 7). The average precipitation in the forest is generally higher than that in the grassland, with a change rate of 106.57 mm/10a for the forest and 90.10 mm/10a for the grassland (Figure 7).

The average ground temperature in the forest is significantly lower than that in the grassland, indicating that forests have a better cooling effect compared to grasslands. Moreover, the average precipitation in the forest is slightly higher than that in the grassland, indicating that forests have a stronger ability to retain moisture.

## 4. Discussion

### 4.1. Vegetation Phenological Change Features in the Horqin Sandy Land

As can be seen from the statistical results in Figure 3, the start date of the vegetation phenological period in the Horqin Sandy Land exhibits significant interannual fluctuations with a slight delay trend, while the end date shows a noticeable delay trend, leading to an overall extension of the phenological period.

On average, in the Horqin Sandy Land, the start date of the phenological period for forested areas is slightly later than that of grasslands, but the end date of the phenological period for forested areas is significantly later than that of grasslands, resulting in a longer phenological period for forests. This difference may be related to the shorter physiological activity of grasslands compared to forests. It indicates that with the increase in temperature, the growth trend of vegetation in forested areas is better than that in grasslands (Figure 8).

### 4.2. Response of Vegetation Phenology to Climatic Factors in Horqin Sandy Land

The statistics in Section 3.2 show that, statistically, the January temperature of the current year has the greatest effect on the start date of the phenological period. Overall, there is a positive correlation between temperature and the start date, indicating that higher temperatures lead to a delayed start date. However, the influence is not significant, suggesting that temperature is not the main influencing factor for the start date of the phenological period. The correlation of the start date with precipitation is stronger than that with temperature, mainly reflected in the delaying effect of cumulative precipitation on the start date. A decrease in cumulative precipitation leads to a delayed start date, indicating that precipitation is an important influencing factor for the start date of the phenological period.

The overall correlation between temperature and the end date of the phenological period is not strong. Except for a noticeable positive correlation with August temperature, the temperatures in other months show weak negative correlations with the end date. This indicates that the impact of temperature on the end date in the Horqin Sandy Land is mainly related to the delaying effect of high temperatures in the summer. This is because higher temperatures favor the extension of the growing season by slowing down the dormancy of trees. The correlation of the end date with precipitation is stronger than with temperature and is generally positive, indicating that an increase in precipitation significantly delays the end date of the phenological period. Precipitation is the primary meteorological factor causing the delay in the end date, which is consistent with previous studies.

The strongest correlation between the end date and precipitation occurs with the precipitation in August, suggesting that the impact of precipitation on the end date is mainly concentrated in this month. The reason might be that the Horqin Sandy Land is located in a typical semi-humid and semi-arid region, and water availability is a limiting factor for vegetation growth. An increase in precipitation favors better vegetation growth.

### 4.3. Change Features in Meteorological Factors in the Horqin Sandy Land

From 2000 to 2021, both the annual average temperature and annual precipitation in the Horqin Sandy Land showed a significant upward trend. This may be related to the implementation of national policies such as returning farmland to forests and grasslands and protecting natural forests after 2000. In the past two decades above, the substantial increase in artificial vegetation has promoted ecological restoration in the sandy region, leading to an overall increase in vegetation cover in the Horqin Sandy Land, The average annual NDVI value increased from 0.187 in 2000 to 0.194 in 2021 (Table 4). There was a significant correlation between the average annual NDVI and the average annual temperature and annual precipitation (Table 5). So, we speculate that changes in vegetation cover alter the exchange of energy, water, and momentum between the land and the atmosphere, thus influencing climate change. Different vegetation types exhibit variations in surface reflectance, evapotranspiration, and surface roughness, resulting in varying degrees of impact on the climate.

The changing climate, in turn, interacts with vegetation through the exchange of matter and energy between the atmosphere and the plants. Previous studies have shown that the rising temperatures in the northern hemisphere can affect vegetation phenology, leading to earlier phenological start dates. Different vegetation types respond differently to climate change, and climate change can promote the growth of forest vegetation and increase forest productivity. This is manifested by an earlier start and a delayed end of the phenological period, as well as an overall lengthening of the phenological period when the average temperature increases. However, higher temperatures can have adverse effects on grassland vegetation. Increased temperatures lead to higher evaporation rates, resulting in a warmer and drier climate in northern China’s pastoral areas. As a consequence, the boundaries of various grassland types will shift eastward, and grassland vegetation in Inner Mongolia will be compressed from southeast to northwest.

## 5. Conclusions

This study conducted a long time-series analysis of the Horqin Sandy Land from 2000 to 2021 using remote sensing monitoring. The following conclusions were reached:Significant changes in vegetation phenology were observed in the Horqin Sandy Land. Both the phenological start date and end dates showed a delayed trend, with change rates of 0.82 days/10a and 5.82 days/10a, respectively. The total phenological period showed an extended trend, with a change rate of 5.00 days/10a, primarily due to the delayed end date. Additionally, the average phenological start date in the sandy land’s forests was around the 129th day, slightly later than the grasslands’ average of 128.68th day. The average phenological end date in the forests was around 286.95th days, later than the grasslands’ average of 283.68th days, resulting in a longer phenological period in forests compared to grasslands.The partial correlation between precipitation and phenological factors was generally higher than the partial correlation between temperature and phenological factors in the sandy land. This indicates that precipitation is the primary influencing factor for changes in vegetation phenology. However, there was little difference in the partial correlation between precipitation and phenological factors in forests and grasslands. The partial correlation between temperature and the phenological start date in grasslands was greater than the partial correlation between temperature and the phenological start date in forests. Similarly, the partial correlation between temperature and the phenological end date in forests was greater than the partial correlation between temperature and the phenological end date in grasslands, suggesting that temperature is the main driver of phenological differences between forests and grasslands.Within the sandy land, the annual mean NDVI value, precipitation, and temperature increased at rates of 0.01/10a, 93.16 mm/10a, and 0.29 °C/10a, respectively. This indicates that while the climate is warming, the ecological environment in the Horqin Sandy Land is gradually improving due to artificial restoration efforts. The rate of increase in forest ground temperature (0.31 °C/10a) was higher than that in grassland ground temperature (0.21 °C/10a), yet the average ground temperature in grasslands was significantly higher than in forests. The rate of increase in precipitation in forests (106.57 mm/10a) was higher than in grasslands (90.10 mm/10a), and the average precipitation in forests was significantly higher than in grasslands, suggesting that forests have better cooling effects and stronger water holding capacity.

## Figures and Tables

**Figure 1 plants-12-03913-f001:**
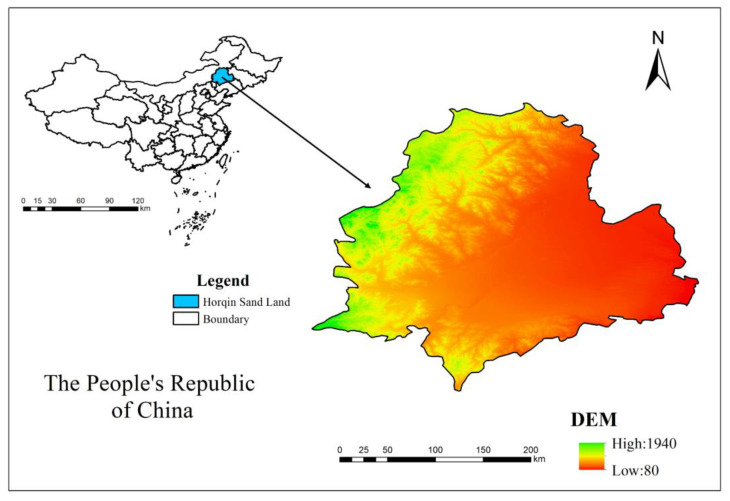
The geographic location of Horqin Sandy Land.

**Figure 2 plants-12-03913-f002:**
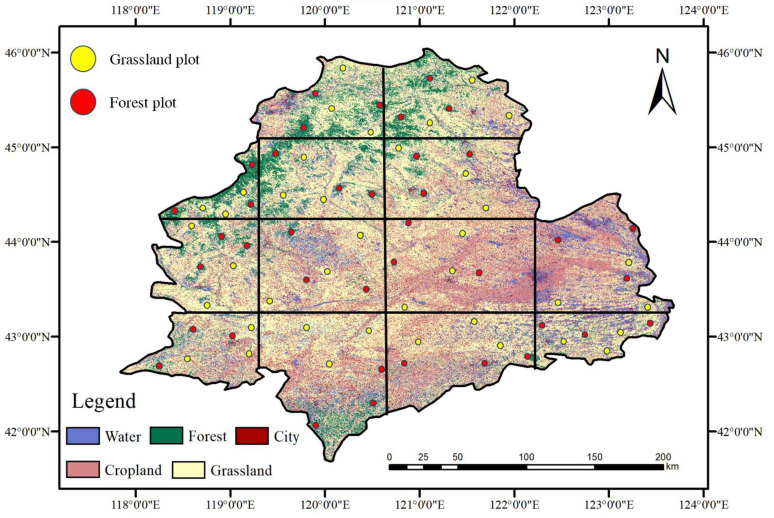
Location of ecological restoration project area.

**Figure 3 plants-12-03913-f003:**
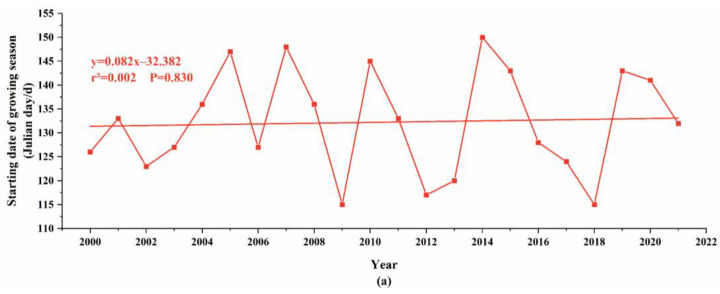
Inter-annual variation of vegetation phenology in Horqin Sandy Land from 2000 to 2021. (**a**) shows inter-annual variation of the beginning of the growing season in Horqin Sandy Land from 2000 to 2021; (**b**) shows inter-annual variation at the end of the growing season in Horqin Sandy Land from 2000 to 2021; (**c**) shows inter-annual variation of growing season length in Horqin Sandy Land from 2000 to 2021.

**Figure 4 plants-12-03913-f004:**
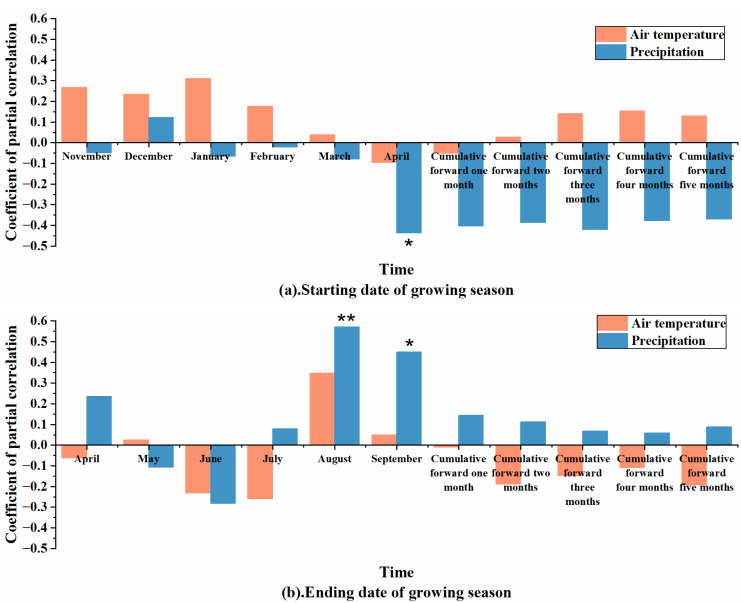
Relationship between vegetation phenology and meteorological factors in Horqin Sandy Land. (**a**) shows a relationship between vegetation phenology starting period and corresponding meteorological factors in Horqin Sandy Land; (**b**) shows the relationship between vegetation phenology ending period and corresponding meteorological factors in Horqin Sandy Land. Note: * means passing the significance test at the 0.05 level, and ** means passing the significance test at the 0.01 level.

**Figure 5 plants-12-03913-f005:**
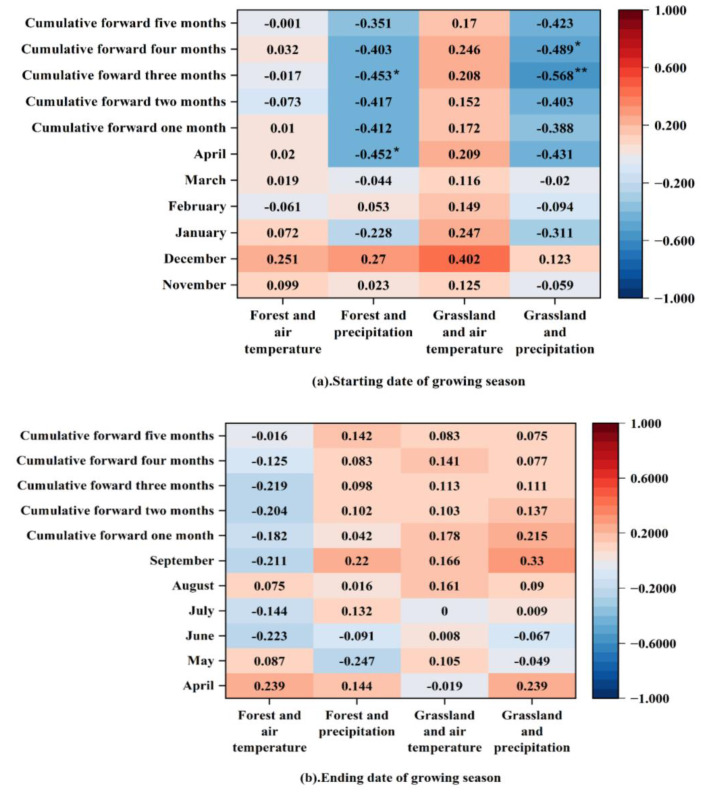
Relationships between forest and grassland phenology and meteorological factors in Horqin Sandy Land. (**a**) shows relationships between forest and grassland phenology starting periods and corresponding meteorological factors in Horqin Sandy Land; (**b**) shows relationships between forest and grassland phenology ending periods and corresponding meteorological factors in Horqin Sandy Land. Note: * means passing the significance test at the 0.05 level, and ** means passing the significance test at the 0.01 level.

**Figure 6 plants-12-03913-f006:**
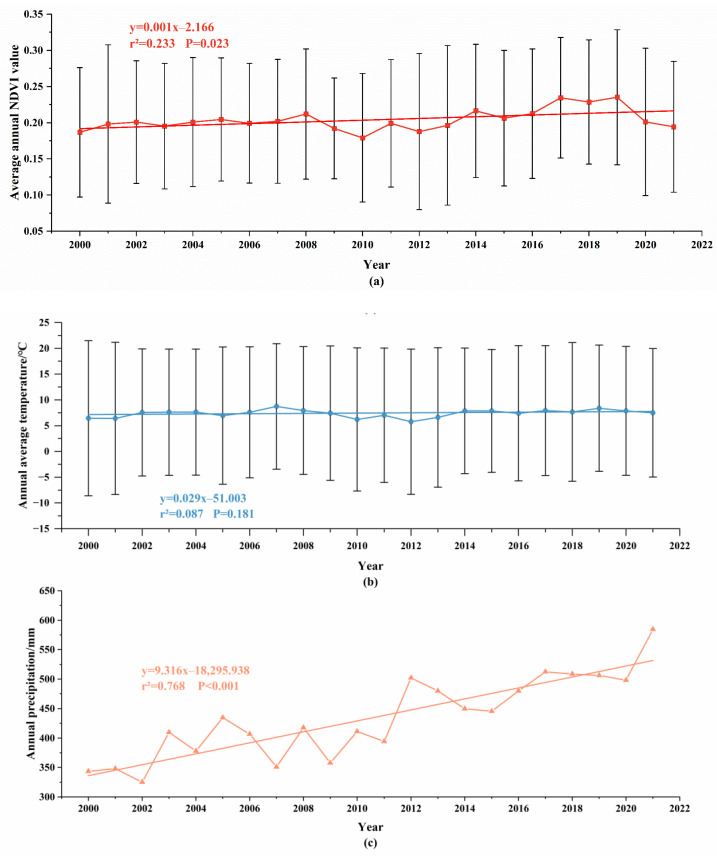
Inter-annual variations of average NDVI, average temperature, and precipitation in Horqin Sandy Land from 2000 to 2021. (**a**) shows inter-annual variations of average NDVI value in Horqin Sandy Land from 2000 to 2021; (**b**) shows inter-annual variations of average temperature in Horqin Sandy Land from 2000 to 2021; (**c**) shows inter-annual variations of precipitation in Horqin Sandy Land from 2000 to 2021.

**Figure 7 plants-12-03913-f007:**
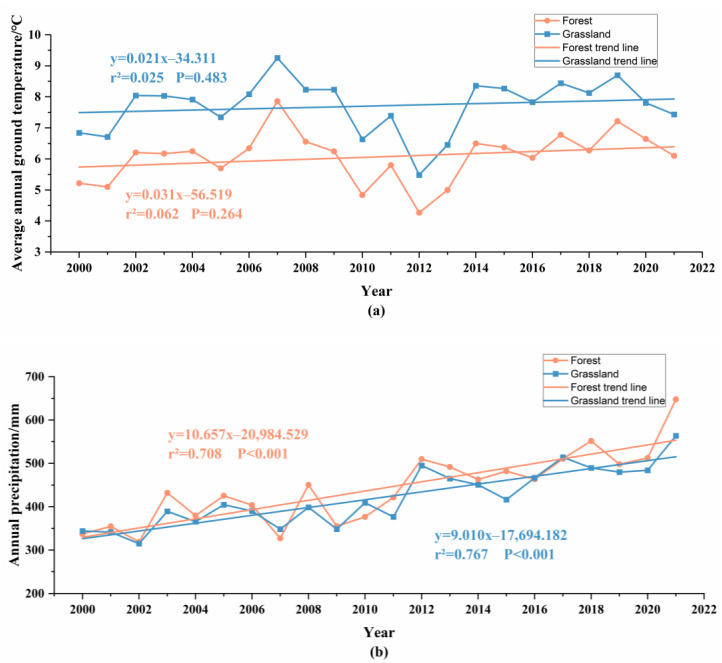
Inter-annual variation of average ground temperature and precipitation of forest and grassland in Horqin Sandy Land from 2000 to 2021. (**a**) shows inter-annual variations of average ground temperature of forest and grassland in Horqin Sandy Land from 2000 to 2021; (**b**) shows inter-annual variations of precipitation of forest and grassland in Horqin Sandy Land from 2000 to 2021.

**Figure 8 plants-12-03913-f008:**
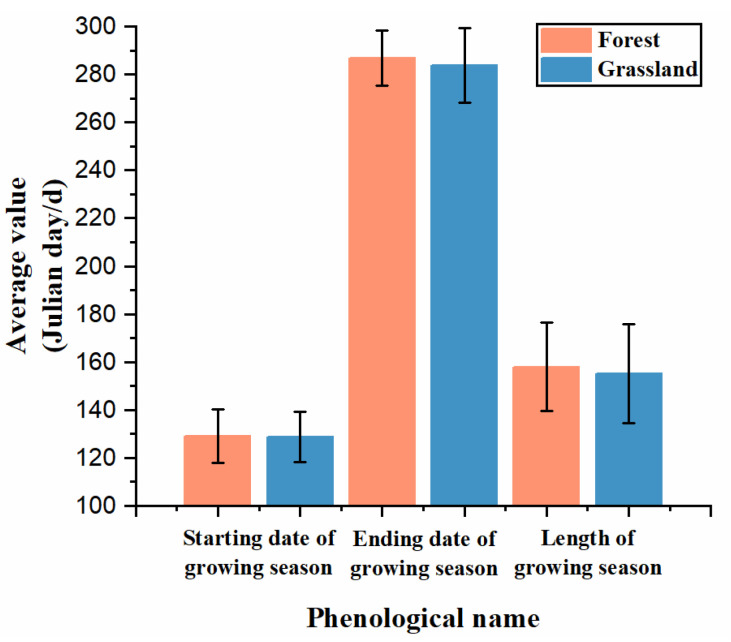
Average values of forest and grassland phenology in Horqin Sandy Land from 2000 to 2021.

**Table 1 plants-12-03913-t001:** Data sources and resolutions.

Data Type	Dataset	Spatial Resolution	Length of Time
NDVI	MODIS/MOD09GA_006_NDVI	500 m	1 d
Precipitation	UCSB-CHG/CHIRPS/DAILY	1 km	1 d
Temperature	ECMWF/ERA5_LAND/DAILY_RAW	1 km	1 d
DEM	NASA/NASADEM_HGT/001	30 m	1 d

**Table 2 plants-12-03913-t002:** Average ground temperature statistics of forest and grassland in Horqin Sandy Land from 2000 to 2021.

Year	Average Annual Ground Temperature of Forest/°C	Ground Temperature Variation/°C	Rate of Change/%	Average Annual Ground Temperature of Grassland/°C	Ground Temperature Variation/°C	Rate of Change/%
2000	5.22	—	—	6.84	—	—
2001	5.10	–0.12	–0.02	6.71	–0.13	–0.02
2002	6.21	1.11	0.22	8.04	1.33	0.20
2003	6.17	–0.04	–0.01	8.04	–0.01	0.00
2004	6.25	0.08	0.01	7.92	–0.12	–0.01
2005	5.70	–0.55	–0.09	7.34	–0.57	–0.07
2006	6.34	0.65	0.11	8.09	0.74	0.10
2007	7.86	1.52	0.24	9.25	1.17	0.14
2008	6.55	–1.31	–0.17	8.23	–1.02	–0.11
2009	6.24	–0.31	–0.05	8.23	0.00	0.00
2010	4.84	–1.41	–0.23	6.63	–1.60	–0.19
2011	5.80	0.96	0.20	7.39	0.76	0.11
2012	4.27	–1.53	–0.26	5.48	–1.91	–0.26
2013	5.00	0.73	0.17	6.45	0.97	0.18
2014	6.50	1.50	0.30	8.36	1.91	0.30
2015	6.37	–0.12	–0.02	8.27	–0.09	–0.01
2016	6.03	–0.34	–0.05	7.83	–0.44	–0.05
2017	6.77	0.74	0.12	8.44	0.61	0.08
2018	6.27	–0.50	–0.07	8.13	–0.31	–0.04
2019	7.21	0.94	0.15	8.70	0.57	0.07
2020	6.65	–0.57	–0.08	7.81	–0.89	–0.10
2021	6.10	–0.55	–0.08	7.44	–0.37	–0.05

**Table 3 plants-12-03913-t003:** Precipitation statistics of forest and grassland in Horqin Sandy Land from 2000 to 2021.

Year	Average Annual Precipitation of Forest/°C	Precipitation Variation/°C	Rate of Change/%	Average Annual Precipitation of Grassland/°C	Precipitation Variation/°C	Rate of Change/%
2000	337.25	—	—	343.83	—	—
2001	354.71	17.46	0.05	341.21	–2.62	–0.01
2002	318.43	–36.28	–0.10	314.64	–26.57	–0.08
2003	432.10	113.68	0.36	389.33	74.69	0.24
2004	379.58	–52.52	–0.12	365.35	–23.99	–0.06
2005	425.13	45.55	0.12	404.62	39.27	0.11
2006	403.61	–21.53	–0.05	389.95	–14.67	–0.04
2007	326.88	–76.72	–0.19	348.22	–41.73	–0.11
2008	450.31	123.43	0.38	398.47	50.25	0.14
2009	354.91	–95.40	–0.21	348.18	50.29	–0.13
2010	376.43	21.52	0.06	409.05	–60.87	0.17
2011	421.23	44.80	0.12	376.74	–32.31	–0.08
2012	509.70	88.47	0.21	495.20	118.46	0.31
2013	491.87	–17.83	–0.03	464.99	–30.21	–0.06
2014	463.00	–28.87	–0.06	450.43	–14.56	–0.03
2015	482.08	19.08	0.04	416.46	–33.97	–0.08
2016	463.47	–18.61	–0.04	466.70	50.24	0.12
2017	510.91	47.44	0.10	514.24	47.54	0.10
2018	551.70	40.80	0.08	489.60	–24.64	–0.05
2019	497.93	–53.77	–0.10	479.94	–9.66	–0.02
2020	512.52	14.59	0.03	483.93	3.99	0.01
2021	647.94	135.42	0.26	563.70	79.76	0.16

**Table 4 plants-12-03913-t004:** Annual average NDVI value statistics of Horqin Sandy Land from 2000 to 2021.

Year	Average Annual NDVI Value	NDVI Value Variation	Rate of Change/%
2000	0.187	—	—
2001	0.198	0.012	0.062
2002	0.201	0.003	0.013
2003	0.195	–0.006	–0.028
2004	0.201	0.006	0.029
2005	0.205	0.004	0.018
2006	0.199	–0.005	–0.026
2007	0.202	0.003	0.014
2008	0.212	0.010	0.049
2009	0.192	–0.020	–0.094
2010	0.179	–0.013	–0.068
2011	0.199	0.020	0.112
2012	0.188	–0.011	–0.057
2013	0.196	0.009	0.047
2014	0.216	0.020	0.101
2015	0.206	–0.010	–0.046
2016	0.212	0.006	0.030
2017	0.234	0.022	0.103
2018	0.229	–0.006	–0.025
2019	0.235	0.007	0.029
2020	0.201	–0.034	–0.145
2021	0.194	–0.007	–0.033

**Table 5 plants-12-03913-t005:** Partial correlation analysis of annual average NDVI value with annual average temperature and annual precipitation in Horqin Sandy Land from 2000 to 2021.

Name	Annual Average Temperature	Annual Precipitation
Annual average NDVI value	0.657 **	0.458 *

Note: * means passing the significance test at the 0.05 level, and ** means passing the significance test at the 0.01 level.

## Data Availability

All the data are available from the corresponding author upon reasonable request.

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
