# Peer review of "Effects of Ecological Restoration and Climate Change on Herbaceous and Arboreal Phenology"

_plants, 2023, doi:10.3390/plants12223913_

Round 1

Reviewer 1 Report

Comments and Suggestions for Authors

Author Response

We thank Editor and an anonymous reviewer for their comments. Based on your comment and request, we have made extensive modification on the original manuscript. We have improved the organization of the manuscript and the presentation of the data, with a focus on the statistical results of the figures. A document answering every question from the referees is also included. The manuscript has been significantly improved by addressing the comments. The following are our point-to-point responses to their comments.

In the abstract, briefly state where the Hoaquin Sandy Area is.

Response: Thank you for your suggestion. I believe you are referring to the "Horqin Sandy Land." Considering this, we have included the following statement in the abstract to provide clarity on its location: "Horqin Sandy Land is situated in close proximity to the eastern part of the West Liaohe River. It is the largest sandy land region in China, characterized by its fragile ecological environment." This addition should help readers understand the geographical context of the study.

Unless there is a specific and compelling reason to do so, the Results section should be separate from the Discussion section. Results are the factual outcomes, whereas the Discussion is interpretation.

Response: Thank you for incorporating the suggestions. We have now organized our manuscript with the results as the third section and the discussion as the fourth section.

The font size on the axis labels needs to be bigger. As is, they are very hard to read.

Response: Implemented. Based on your suggestions, we have adjusted the font size of the axis labels. Specifically, we have increased the font size of the axis headings in Figure 3, 4, 6, 7, and 8 from 11 to 14, and the font size of the axis labels from 11 to 12. We believe that these changes have improved the clarity and readability of the figures, making them easier to understand.

All figures and tables also need statistical results. As appropriate to the test that was conducted, provide p-values, r2 values, etc.

Response: Implemented. We made changes in these two aspects as following.

  1. In Figure 3, 6, and 7, we carried out the result statistics and added the display of P values and r² values to the figure to make the data more complete.
  2. In Figure 4 and 5, We conducted a P test on the data and marked it on the figure. Thereof, * means passing the significance test at the 0.05 level and ** means passing the significance test at the 0.01 level.

On Figure 3, move the x-axis to -0.5 so that it can be read.

Response: Implemented. According to your advice, we moved the x-axis to -0.5 on Figure 3. But with adjustments, it is now changed from Figure 3 to Figure 4. Maybe it's easier to read now.

In Figure 4, be specific about what the numbers are. Perhaps they are correlation coefficients? We need p-values, too.

Response: Implemented. The numbers in Figure 4 are exactly correlation coefficients. We conducted a P test on these data and marked it on the figure. Thereof, * means passing the significance test at the 0.05 level and ** means passing the significance test at the 0.01 level.

The Materials and Methods section needs to be between the Introduction and the Results.

Response: Implemented. We adjusted the position of Materials and Methods. Now the Material and Method is the second part of the article, the Results is the third part.

Reviewer 2 Report

Comments and Suggestions for Authors

Page 2, line 82. "A research indicates". Reword. Suggest "Research indicates".

Page 4, line 130. "shorter successional time of grassland". This is a misuse of "successional" which implies change of community species composition over time. It appears here that you are indicating that forests are physiologically active longer than grasslands.

Page 7, lines 211-220. You have shown changes in phenology over 21 years related to increasing temperature and precipitation. In the introduction you mention restoration; however, you do not quantify amount of vegetation restoration or present any empirical basis for concluding that the phenology, temperature, and precipitation changes are related to restoration. Unless you can make that case in this paper, the claim should be removed.

Page 12, lines 285-288. Italicize scientific names.

Page 14, lines 328-330. Did you derive NVDI from the MODIS imagery? Need to specify where it came from.

Page 12, lines 338 and 340. Do you mean NVDI ratio? Not radio?

Pages 16-18, References. Remove unneeded spaces in Qinghai Tibetan Plateau in # 3. Italicize scientific names in #5, #8, and #23. Consistently capitalize journal titles in #8, #16, #25, #27, and #31.

Comments on the Quality of English Language

See above

Author Response

We thank Editor and an anonymous reviewer for their comments. Based on your comment and request, we have made extensive modification on the original manuscript. We have improved some aspects of language use, spelling and formatting. A document answering every question from the referees is also included. The manuscript has been significantly improved by addressing the comments. The following are our point-to-point responses to their comments.

Page 2, line 82. "A research indicates". Reword. Suggest "Research indicates".

Response: Implemented. According to your suggestions, we changed "A research indicates" to "Research indicates". Please see Page 2, line 85.

Page 4, line 130. "shorter successional time of grassland". This is a misuse of "successional" which implies change of community species composition over time. It appears here that you are indicating that forests are physiologically active longer than grasslands.

Response: Implemented. We appreciate your attention to the issue we highlighted. We have now revised the terminology from "successional" to "physiological active." Please refer to Page 14, line 333 for this update.

Page 7, lines 211-220. You have shown changes in phenology over 21 years related to increasing temperature and precipitation. In the introduction you mention restoration; however, you do not quantify amount of vegetation restoration or present any empirical basis for concluding that the phenology, temperature, and precipitation changes are related to restoration. Unless you can make that case in this paper, the claim should be removed.

Response: Implemented. Thank you for your suggestions. We added Table 3 and Table 4 to quantify amount of vegetation restoration and present that temperature and precipitation changes are related to restoration. Please see Page 16, line 381-386.

Page 12, lines 285-288. Italicize scientific names.

Response: Implemented. We've italicized all the scientific names. Please see Page 3, line 127-130.

Page 14, lines 328-330. Did you derive NVDI from the MODIS imagery? Need to specify where it came from.

Response: Implemented. We exactly derived NDVI from the MODIS imagery. So we added “Based on MODIS images, the NDVI values of each day are calculated on the platform, and the data is denoised by S-G filtering” in the article, so that we can know where the NDVI values came from. Please see Page 4, line 150-152.

Page 12, lines 338 and 340. Do you mean NVDI ratio? Not radio?

Response: Implemented. We are very sorry for this and we have corrected it. Please see  Page 5, line 177-178.

Pages 16-18, References. Remove unneeded spaces in Qinghai Tibetan Plateau in # 3. Italicize scientific names in #5, #8, and #23. Consistently capitalize journal titles in #8, #16, #25, #27, and #31.

Response: Implemented. We made changes references in these three areas:

  1. We removed unneeded spaces in Qinghai Tibetan Plateau in # 3.
  2. We Italicized scientific names in #5, #8, and #23.
  3. We capitalized journal titles in #8, #16, #25, #27, and #31.

Round 2

Reviewer 2 Report

Comments and Suggestions for Authors

Page 3, line 129. Should read "Based on this, this study ...and uses remote sensing..." Not "and use".

Page 3, line 133. Remove unneeded "changes" to read "the changes of some climate factors."

Page 3, line 155. Should read "The ratio of sand dunes to grassland..."

Page 16, line 438. "NVDI value statistics..." Not valve.

Comments on the Quality of English Language

See above.

Author Response

We thank Editor and an anonymous reviewer for their comments. Based on your comment and request, we have made extensive modification on the original manuscript. We have improved some aspects of language use. A document answering every question from the referees is also included. The manuscript has been significantly improved by addressing the comments. The following are our point-to-point responses to their comments.

Page 3, line 129. Should read "Based on this, this study ...and uses remote sensing..." Not "and use".

Response: Implemented. According to your suggestions, we changed "and use" to "and uses". Please see Page 3, line 99.

Page 3, line 133. Remove unneeded "changes" to read "the changes of some climate factors."

Response: Implemented. Thank you for your suggestions. We have removed unneeded "changes". Please see Page 3, line 103.

Page 3, line 155. Should read "The ratio of sand dunes to grassland..."

Response: Implemented. We have changed the sentence to "The ratio of sand dunes to grassland...". Please see Page 3, line 126.

Page 16, line 438. "NVDI value statistics..." Not valve.

Response: Implemented. We are very sorry for this and we have corrected it. Please see Page 16, line 381.
